# Regulated aggregative multicellularity in a close unicellular relative of metazoa

Arnau Sebé-Pedrós[1,2], Manuel Irimia[3], Javier del Campo[1], Helena Parra-Acero[1], Carsten Russ[4], Chad Nusbaum[4], Benjamin J Blencowe[3]*, Iñaki Ruiz-Trillo[1,2,5]*

[1]Department of Functional Genomics and Evolution, Institut de Biologia Evolutiva (CSIC–Universitat Pompeu Fabra), Barcelona, Spain; [2]Departament de Genètica, Facultat de Biologia, Universitat de Barcelona, Barcelona, Spain; [3]Donnelly Centre, University of Toronto, Toronto, Canada; [4]Genomics Platform, Broad Institute of Harvard and the Massachusetts Institute of Technology, Cambridge, United States; [5]Institució Catalana de Recerca i Estudis Avançats (ICREA), Barcelona, Spain

**Abstract** The evolution of metazoans from their unicellular ancestors was one of the most important events in the history of life. However, the cellular and genetic changes that ultimately led to the evolution of multicellularity are not known. In this study, we describe an aggregative multicellular stage in the protist *Capsaspora owczarzaki*, a close unicellular relative of metazoans. Remarkably, transition to the aggregative stage is associated with significant upregulation of orthologs of genes known to establish multicellularity and tissue architecture in metazoans. We further observe transitions in regulated alternative splicing during the *C. owczarzaki* life cycle, including the deployment of an exon network associated with signaling, a feature of splicing regulation so far only observed in metazoans. Our results reveal the existence of a highly regulated aggregative stage in *C. owczarzaki* and further suggest that features of aggregative behavior in an ancestral protist may had been co-opted to develop some multicellular properties currently seen in metazoans.

**\*For correspondence:**
b.blencowe@utoronto.ca (BJB);
inaki.ruiz@multicellgenome.org
(IR-T)

**Reviewing editor**: Diethard Tautz, Max Planck Institute for Evolutionary Biology, Germany

## Introduction

Living organisms emerged from the integration of multiple levels of organization. These levels were shaped by both physiochemical constraints and historical circumstances, the latter being more important in more complex systems (*Jacob, 1977*). Therefore, it is important to identify the phylogenetic inertia (*sensu Burt, 2001*) imposed by the raw starting material in order to properly understand major evolutionary transitions, such as the origin of metazoan multicellularity (*Knoll, 2011*). Examination of both the genetic repertoire (*King, 2004*; *Ruiz-Trillo et al., 2007*; *Rokas, 2008*) and the cell types present in the immediate unicellular relatives of metazoans can provide insights into this evolutionary transition, as they reveal the historical constraints in early metazoan evolution. In this regard, the analyses of unicellular holozoan genomes, that is choanoflagellates and filastereans, have shown that the genetic repertoire of the metazoan unicellular ancestor was much more complex than previously thought (*Abedin and King, 2008*; *King et al., 2008*; *Sebé-Pedrós et al., 2010*; *Suga et al., 2013*).

Multicellularity has been independently acquired multiple times during the evolution of eukaryotes, in more than 20 different lineages including animals, plants, fungi, slime molds, green and brown algae, and several other eukaryotes (*King, 2004*; *Parfrey and Lahr, 2013*). Multicellular organisms evolved through two major strategies: aggregation of different cells or clonal division of a single cell. Multi-level selection theory has proposed that the most complex multicellular organisms likely arose through clonal development rather than by aggregation of genetically diverse cells, since intra-organismal

**eLife digest** When living things made from many cells evolved from single-celled ancestors, it was a breakthrough in the history of life—and one that has occurred more than once. In fact, multicellular life has evolved independently at least 25 times, in groups as diverse as animals, fungi, plants, slime molds and seaweeds. There are broadly two ways to become multicellular. The most complex multicellular species, such as animals, will replicate a single cell, over and over, without separating the resultant cells. However, in species that are only occasionally multicellular, free-living cells tend instead to join together in one mass of many cells.

Evolution is constrained by its raw materials; so looking at the living relatives of a given species, or group, can lead to a better understanding of its evolution because its relatives contain clues about its ancestors. To gain insights into how animal multicellular life might have began; Sebé-Pedrós et al. studied the life cycle of the amoeboid organism *Capsaspora owczarzaki*. Found within the bodies of freshwater snails, this single-celled amoeba is a close relative of multicellular animals and could resemble one of their earliest ancestors.

At certain stages of the life cycle Sebé-Pedrós et al. noticed that individual amoebae gathered together to form a multicellular mass—something that had not been seen before in such a close relative of the animals. Moreover, the genes that 'switched on' when the amoebae began to aggregate are also found in animals; where, together with other genes, they control development and the formation of tissues. Sebé-Pedrós et al. suggest that the first multicellular animals could have recycled the genes that control the aggregation of single-celled species: in other words, genes that once controlled the changes that happen at different times in a life cycle, now control the changes that develop between different tissues at the same time.

Sebé-Pedrós et al. also observed that alternative splicing—a process that allows different proteins to be made from a single gene—occurs via two different mechanisms during the life cycle of *Capsaspora*. Most of the time *Capsaspora* employs a form of alternative splicing that is often seen in plants and fungi, and only rarely in animals; for the rest of the time it uses a form of alternative splicing similar to that used by animal cells.

The evolution of complex alternative splicing mechanisms is a hallmark feature of multicellular animals. The exploitation of two major forms of alternative splicing in *Capsaspora* could thus reflect an important transition during evolution that resulted in an increased diversity of proteins and in more complex gene regulation. Such a transition may ultimately have paved the way for the increased specialization of cell types seen in animals.

This glimpse into the possible transitions in gene regulation that contributed to the birth of multicellular animals indicates that the single-celled ancestor of the animals was likely more complex than previously thought. Future analyses of the animals' close relatives may further improve our understanding of how single-celled organisms became multicellular animals.

competition in the latter might be expected to be evolutionarily unstable (*Grosberg and Strathmann, 2007*; *Michod, 2007*; *Newman, 2012*). Accordingly, eukaryotic lineages that attained the most complex multicellular lifestyles (i.e., plants and metazoans) arose through clonal cell division (*Grosberg and Strathmann, 2007*). In contrast to clonal multicellularity, aggregative cell behavior typically represents a transient life cycle stage. This type of multicellularity arose within several eukaryotic clades, including the dictyostelids (Amoebozoa) (*Schaap, 2011*), acrasid amoebas (Heterolobosea, Discicristata, Discoba) (*Brown et al., 2011*), *Guttulinopsis vulgaris* (Cercozoa, Rhizaria) (*Brown et al., 2012*), the genus *Sorogena* (Ciliata, Alveolata) (*Lasek-Nesselquist and Katz, 2001*), the holomycota *Fonticula alba* (Opisthokonta) (*Brown et al., 2009*) and the genus *Sorodiplophrys* (Labyrinthulomycetes, Heterokonta) (*Dykstra and Olive, 1975*) (*Figure 1*).

Within the opisthokont clade that comprises Metazoa, Fungi and their unicellular relatives (*Cavalier-Smith, 2003*; *Steenkamp et al., 2006*; *Ruiz-Trillo et al., 2008*), so far only a single taxon has been described to have aggregative behavior, which is *F. alba* (*Brown et al., 2009*), a close relative of Fungi. Moreover, among close unicellular relatives of Metazoa, clonal development is the only known multicellular behavior, as described in choanoflagellates and ichthyosporeans (*Dayel et al., 2011*; *Suga and Ruiz-Trillo, 2013*). Within metazoans, which have largely clonal development, some cells show

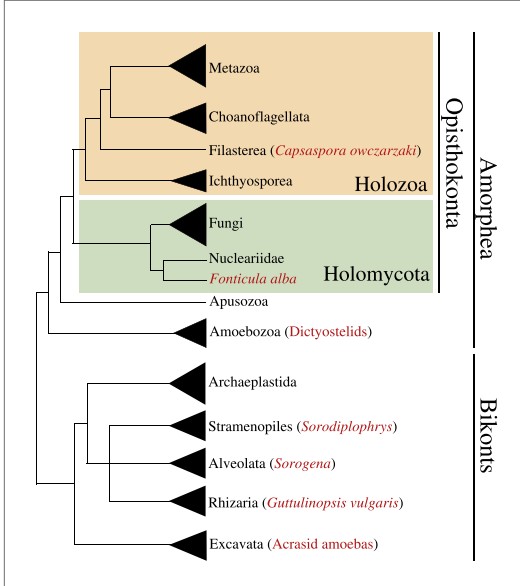

**Figure 1**. Phylogenetic position of *Capsaspora owzarzaki* within the eukaryotes. The Holozoa clade (in yellow) includes Metazoa and their closest unicellular relatives: Choanoflagellata, Filasterea, and Ichthyosporea. *C. owczarzaki* represents one of the two known filastereans taxa that form the sister-group of choanoflagellates and metazoans. Other major eukaryotic groups are shown. Within each group, species or clades with aggregative multicellularity (see text) are highlighted in red.

aggregative behaviors; for example, mesenchymal (*O'Shea, 1987*) and germ line cells (*Savage and Danilchik, 1993*) during development, sponge cells after cell dissociation (*Wilson, 1907*) and arthropod blood cells through active amoeboid movement (*Loeb, 1903*, *1921*).

To gain deeper insight into the possible transitions that arose during the emergence of metazoan multicellularity, we have performed a detailed examination of the life cycle and associated transcriptomic changes of *Capsaspora owczarzaki*, one of the closest known unicellular relatives of metazoans (*Figure 1*). Isolated decades ago as an endosymbiont of the fresh-water snail *Biomphalaria glabrata* (*Owczarzak et al., 1980*), *C. owczarzaki* belongs to the clade Filasterea, the sister-group of Metazoa and choanoflagellates (*Torruella et al., 2012*). Filasterea also includes a free-living marine unicellular species known as *Ministeria vibrans* (*Shalchian-Tabrizi et al., 2008*).

We analyzed the *C. owczarzaki* life cycle and its regulation using electron microscopy, flow cytometry and high-throughput RNA sequencing (RNA-Seq). Through these analyses, we show that *C. owczarzaki* life cycle is tightly regulated at the level of gene expression and alternative splicing (AS). Moreover, we demonstrate the existence of an aggregative multicellular stage in *C. owczarzaki* in which many orthologs of genes important for metazoan clonal multicellularity are upregulated.

## Results and discussion

Under initial culture conditions ('Materials and methods'), *C. owczarzaki* differentiates into an amoeba that crawls over substrate (*Video 1*), surveying its environment with its filopodia (*Sebé-Pedrós et al., 2013*). At this stage, active DNA replication occurs (with >10% of the cells in S-phase) and, within 48 hr, the cells enter an exponential growth phase (*Figures 2 and 3*). Subsequently, the cells start to detach from the surface and begin to retract their filopodia and encyst (*Figure 3B,C*). After 8 days, no attached amoebas remain and growth is stabilized (*Figures 2 and 3*). This cystic stage may represent a dispersal resistance form. Strikingly, we observe an alternative path to this process involving the active formation of cell aggregates (*Videos 2 and 3*). In these aggregates, the cells attach to each other and produce cohesive extracellular material (*Figure 3D*) until a compact cell aggregate, in which cells no longer bear filopodia, is formed (*Figure 3E*). Transmission electron microscopy demonstrates the presence of a thick, unstructured, extracellular material within the aggregates that appears to prevent direct contact between cells (*Figure 3F*). Clusters of cells appear to occur at random under normal culture conditions.

To investigate whether *C. owczarzaki* cell clusters are formed through aggregation or clonal division, we first mixed two differentially stained populations of cells ('Materials and methods') and induced aggregate formation, resulting in dual-colored cell clusters (*Figure 4A*). This indicates that cell clusters are not composed of daughter cells resulting from successive cell divisions, which would result in single-color cell clusters, but instead by aggregation of multiple cells. We also observed that aggregates could form efficiently even when cell division was blocked by two different inhibitors, hydroxyurea and aphidicolin (*Figure 4B*). Finally, by flow cytometry, we observed that the proliferation rate of aggregative cells (*Figure 4—figure supplement 1*) is extremely low (compared with the observations in *Figure 2*). Overall, these results show that *C. owczarzaki* cell clusters form by active cell aggregation, not by clonal cell division. This observation represents the first reported case of aggregative multicellularity in a close unicellular relative of Metazoa.

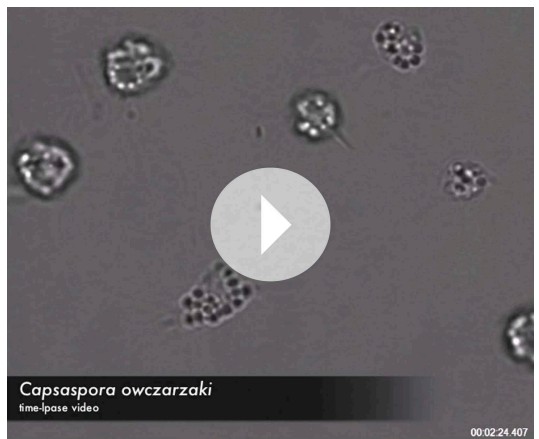

**Video 1**. *C. owczarzaki* filopodial amoeba stage cells crawling. Dark and refringent vesicles can be observed inside each cell. Up to nine different cells can be observed in the video. Also available on YouTube: http://youtu.be/0Uyhor_nDts.

The aggregative multicellularity observed in *C. owczarzaki* adds an additional cell behavior to those already known among extant close unicellular relatives of Metazoa (i.e., choanoflagellates, ichthyosporeans, and filastereans [*Torruella et al., 2012*]). We can then infer that multiple cell types and behaviors (including aggregative behavior, flagellar motility, amoeboid movement, clonal colony formation, etc) were most likely present among the unicellular ancestors of metazoans. This range of cell behaviors may have provided the basis for the evolution of the diverse cell types seen in animals (*Arendt, 2008*; *Arendt et al., 2009*). Interestingly, each one of the three known unicellular lineages closely related to Metazoa (choanoflagellates, ichthyosporeans and filastereans) has some kind of simple multicellularity. Moreover, the tight regulation observed in *C. owczarzaki* (see below) emphasizes that a regulated temporal cell differentiation was already in place among unicellular ancestor of animals. This reinforces the view that, during the transition to animal multicellularity, ancestral premetazoan cell types may have been integrated into a single multicellular entity by means of controlling cell differentiation spatially, rather than temporally (*Mikhailov et al., 2009*). An alternative explanation is that some of the cell behaviors observed in extant unicellular relatives of Metazoa may had evolved independently in some particular unicellular holozoan lineages or species, and do not represent ancestral states. The limited taxon sampling in many of these poorly studied lineages makes it difficult to reliably assess whether these are derived or ancestral characters and this situation is especially dramatic in the case of filastereans, in which only two species have been described so far (*C. owczarzaki* and the free-living *M. vibrans*).

To investigate the molecular composition and regulation of the distinct life cycle stages of *C. owczarzaki*, we isolated filopodiated amoebae, aggregates, and cysts, and analyzed their transcriptomes using RNA-Seq. Of 8637 annotated genes, 4486 showed statistically significant differential regulation ('Materials and methods') in one or more pair-wise comparisons between life cycle stages, including 1354 changes between filopodial and aggregate stages, 3227 between filopodial and cystic stages, and 3096 between aggregate and cystic stages. Moreover, when performing one-versus-all comparisons, each cell stage had a specific transcriptomic profile (*Figure 5A*), indicating tight regulation at the level of gene expression. Using both pairwise and one-versus-all comparisons, we identified significantly enriched gene ontology (GO) categories (*Figures 5 and 6*) and Pfam protein domains (*Figure 7*) in each set of differentially expressed (both up and down-regulated) genes (p<0.01 for each significant category; Fisher's exact test). Genes upregulated in the filopodial stage were enriched in signalling functions, such as tyrosine kinase activity and G-protein-coupled receptor activity, as well as in transcription factors, especially of the Basic Leucine Zipper Domain (bZIP) superfamily (*Figures 5 and 6*). Genes involved in protein synthesis and DNA replication were also significantly upregulated, consistent with the rapid cell proliferation at this stage observed by flow cytometry (*Figure 2*), and further suggesting a high metabolic rate.

When compared to filopodial and aggregative cells, cystic cells showed significant downregulation of genes associated with myosin transport, translation, DNA replication and metabolic activities (especially mitochondrial energy production). However, genes involved in vesicle transport and autophagy were significantly upregulated at this stage (*Figure 5*). These differences may reflect recycling of intracellular components triggered by starvation or other adverse conditions, as has been observed under conditions of adaptive cell survival in other eukaryotes (*Kiel, 2010*). Protein domains involved in the ubiquitin pathway (e.g., UQ_con, zf-RING2 and Cullin domains) and in synaptic cell–cell communication, such as SNARE, synaptobrevin and syntaxin, as well specific transcription factor families (e.g., bHLH transcription factors), were also significantly upregulated in the cystic cells (*Figure 7*). Altogether, these results suggest that major cytosolic rearrangement and protein turnover occur at the cystic stage.

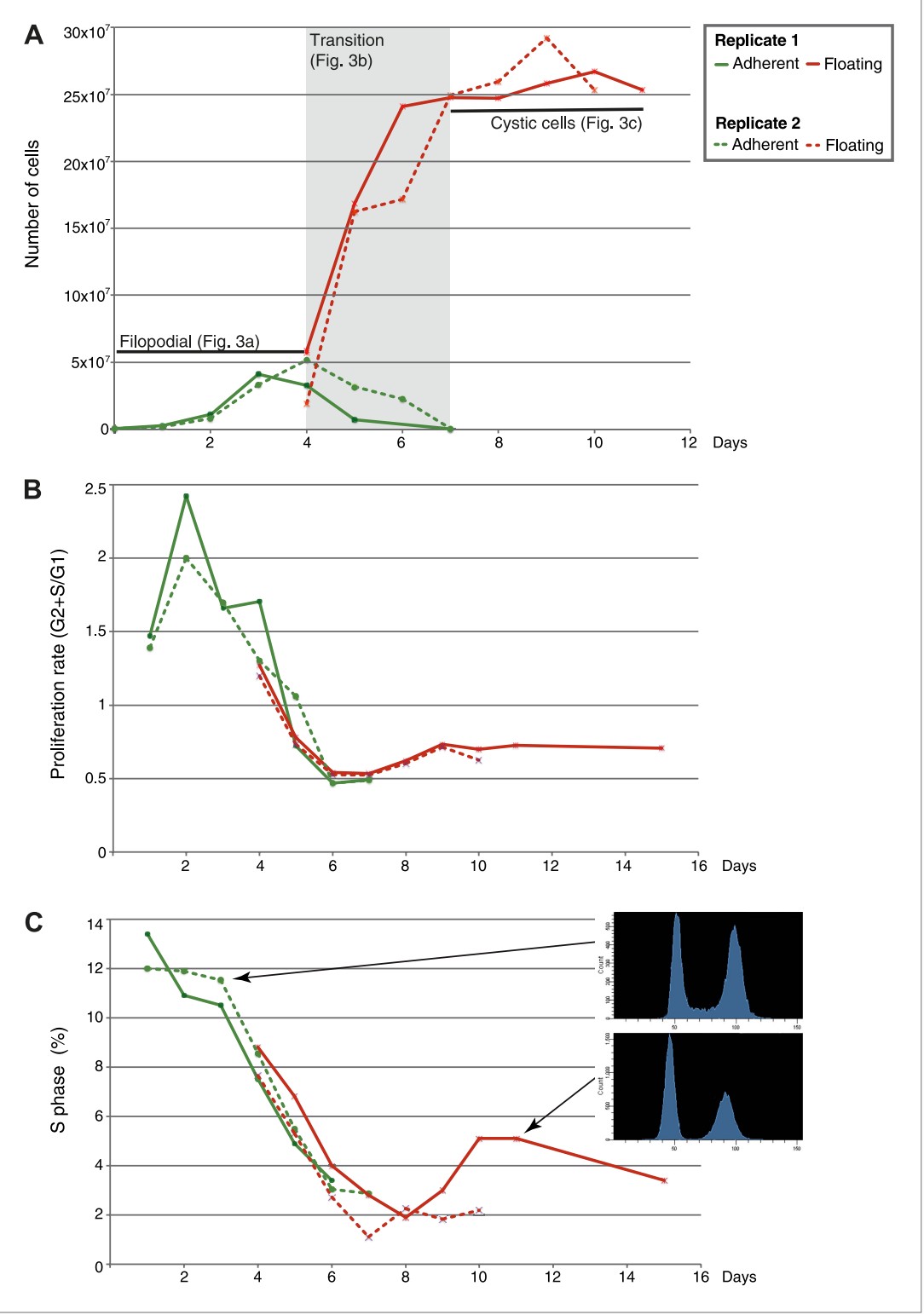

**Figure 2**. Flow-cytometry analysis of *C. owzarzaki* cell cycle. (**A**) Total number of cells per day in each fraction (adherent and floating, see 'Materials and methods'). (**B**) Proliferation rate per day. (**C**) Percentage of cells in S-phase per day and two examples of DNA-content profiles obtained from days 3 and 11. Note the reduction in the number of G2/M cells (second peak) and the drastic reduction in S-phase cells (the area between the two peaks). Data from adherent cells ('Materials and methods') is shown in green and data from floating cells in red. Experimental replicate 1 results are shown with solid lines and replicate 2 results with dashed lines.

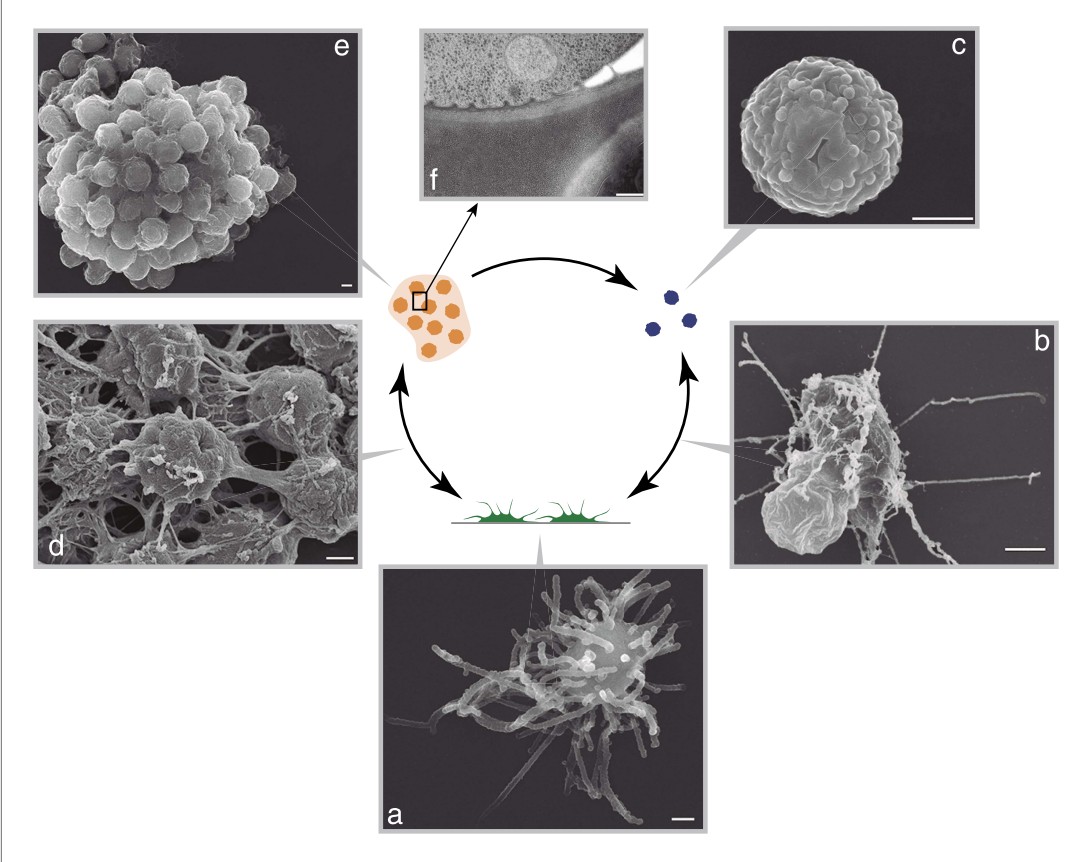

**Figure 3**. *C. owzarzaki* life cycle. (**A**) Filopodial stage cells, amoebas with long filopodia. (**B**) Transition from filopodial to cystic stage: cells retract filopodia. (**C**) Cystic stage cells are rounded cysts without filopodia and slightly smaller than filopodial cells. (**D**) Transition from filopodial to aggregative stage: cells attach to each other and an extracellular matrix appears. (**E**) Mature aggregate. (**F**) Transmission EM showing adjacent cells in the aggregate separated by extracellular matrix. Arrows indicate the observed stage inter-conversions. Scale bars = 1 μm, except in panel **D** = 200 nm.

Remarkably, the aggregative stage showed strong upregulation of the components of the integrin adhesome and associated signalling and cell-adhesion proteins (*Figures 5 and 8A,B*), such as the LamininG domain-containing protein CAOG_07351 (which contains a N-terminal signal peptide sequence and therefore is likely to be secreted) (*Figure 8C*), the IPP complex (ILK-PINCH-Parvin) signalling module, G-protein α-13 (*Gong et al., 2010*), several cytoplasmatic tyrosine kinases (*Hamazaki et al., 1998*) and two receptor tyrosine kinases (which possess extracellular DERM [*Lewandowska et al., 1991*] and fibronectin_3 domains, known to interact with integrins [*Figure 8C*]). These observations strongly suggest that the integrin adhesome and the likely associated tyrosine kinase signalling genes play an important role in the formation of the *C. owzarzaki* aggregates. Furthermore, we also observed upregulation of genes involved in mitosis and in the tubulin cytoskeleton (e.g., kinesins) at the aggregative stage (*Figure 5*). These results indicate that a molecular repertoire associated with animal multicellularity, could function either in aggregative or in clonal multicellularity and in different phylogenetic contexts, in line with previous hypotheses (*Newman, 2012*).

A hallmark feature of the evolution of metazoan multicellularity and cell type diversity is the expansion of AS complexity and regulation through exon skipping, which has entailed the formation of cell type-specific networks of co-regulated exons belonging to functionally related or pathway-specific genes (*Irimia and Blencowe, 2012*). In contrast, differential intron retention is the most widespread form of AS found in non-metazoan eukaryotic species (*McGuire et al., 2008*). To assess the extent to which these forms of AS may contribute to gene regulation in *C. owczarzaki*, we systematically mapped reads from each life cycle stage to a comprehensive set of intron–exon and exon–exon junctions

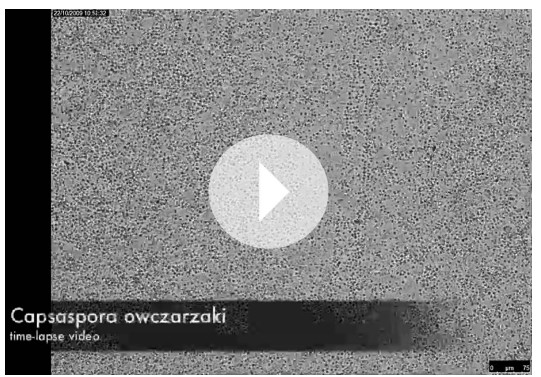

**Video 2**. *C. owczarzaki* cells aggregation. Also available on YouTube: http://youtu.be/83HB8srWQw4.

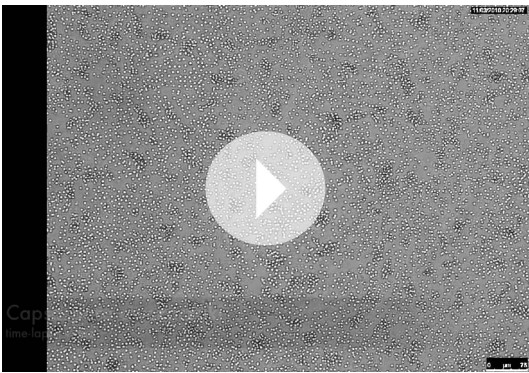

**Video 3**. *C. owczarzaki* cells aggregation. Also available on YouTube: http://youtu.be/OvI6BvBucrc.

(i.e., formed by exon/intron inclusion and skipping) to score their differential usage. Of 25,677 introns with sufficient RNA-Seq read coverage across the three life cycle stages, 2986 (11.6%) showed ≥20% PSI (Percent Spliced In, percent of transcripts from a given gene in which the intron sequence is present) in at least one stage, and approximately a third of genes had at least one such intron retention event. Moreover, we observed marked differences in the extent to which detected intron retention is differentially regulated between the different life cycle stages (***Figure 9A***). In particular, 797 retained introns (in 441 genes) and 259 retained introns (in 232 genes) display differential PSI (dPSI) values of 25% or more in the filopodial and cystic stages compared to the other stages, respectively (***Figure 9B***). In contrast, no retained introns were found to be differentially spliced at the aggregative stage. Most (12 out of 15, 80%) of the analyzed cases of differentially retained introns were validated by RT-PCR (***Figure 9C*** and ***Figure 9— figure supplement 1***).

GO enrichment analysis for the two sets of differentially retained introns showed distinct gene function enrichment (e.g., protein kinase activity and intracellular targeting in the filopodial stage, and histone modification and myosin complex in the cystic stage) (***Figure 6***), implying that regulated intron retention plays different roles at these stages. A low fraction of read-through introns (with length-multiples of three and no in-frame stop codons) were found among the two sets of differentially retained introns, suggesting that most if not all of these retained introns act by reducing the levels of spliced mRNAs exported from the nucleus and translated into protein, as has been observed previously for regulated retained introns in metazoan species (***Yap et al., 2012***). Moreover, we observe that multiple introns belonging to a gene can be retained in a stage-specific manner. For instance, >73% and >29% of multi-intronic genes with one differentially retained intron had at least one additional differentially retained intron at the filopodial or cystic-specific stages, respectively, and 22% and 5% of genes at these stages showed evidence of high retention for all introns in the same genes. Furthermore, RT-PCR analyses and mate information from paired-end read analyses suggested that multiple intron retention events often occur in a combinatorial manner (***Figure 9—figure supplement 1***), thereby increasing the potential impact of intron retentions on mRNA regulation. All of the above observations were highly consistent across three biological replicates (***Figure 9B***), and not observed for neighbouring genes, ruling out contamination of genomic DNA.

We analyzed different features of differentially retained introns that may account for their stage-specific regulation. First, we compared intron lengths. While filopodial-specific differentially retained introns have a similar length distribution to constitutive (PSI less than 2% across all stages) introns, cystic stage-specific introns were significantly longer (p=1.7e$^{-14}$ Wilcoxon rank sum test) (***Figure 9D***). In line with this observation, the average level of intron retention increased steadily with intron length only in the cystic stage (***Figure 9E***). Furthermore, cystic stage-specific retained introns harbored significantly weaker canonical 5′ and 3′ splice site signals than other intron sets (p<0.0013 Wilcoxon rank sum test for all comparisons). Collectively, these data suggest that differential intron retention in the cystic stage may be associated with suboptimal introns (i.e., long and with weak splice sites) that are more efficiently spliced in the other cell stages. In the case of the filopodial-specific differentially

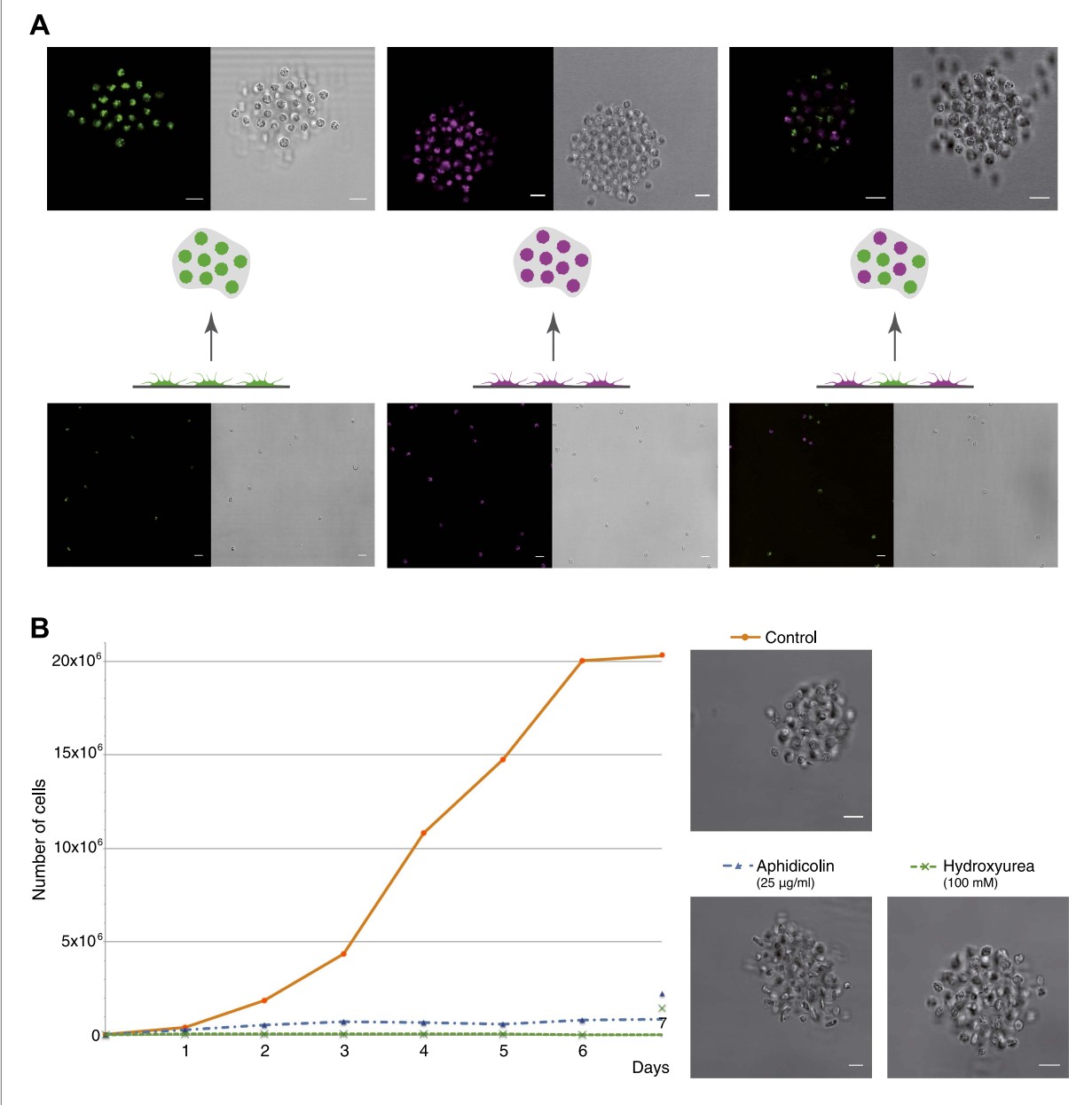

**Figure 4**. *C. owczarzaki* cell clusters form by active aggregation, not clonal cell division. Aggregation was induced in different stained cell populations ('Materials and methods'). (**A**) Left, population of cells stained with Lysotracker (green), uniformly green aggregates are formed. Center, population of cells stained with Chormeo Mitochondrial Staining (cyan), uniformly cyan aggregates are formed. Right, two independently stained populations of cells (green or cyan) are mixed and dual color aggregates are formed, indicating that cells from different origins aggregate to each other. (**B**) Total number of cells per day in control cells, aphidicolin-treated cells and hydroxyurea-treated cells. Note that cell division is blocked by both aphidicolin and hydroxyurea. Aggregate formation was evaluated under each condition. All cells, even those treated with aphidicolin or hydroxyurea, developed aggregates. A representative aggregate is shown for each condition. Scale bars= 10 μm.

The following figure supplements are available for figure 4:

**Figure supplement 1**. Proliferation rate per day of aggregative cells.

retained introns, analyses of sequence motif enrichment with MEME show enrichment of a long T/G-rich motif that highly resembles a recently identified consensus binding site for Elav-like protein in mammals (***Ince-Dunn et al., 2012***) (***Figure 10***). Interestingly, the single ortholog for this gene in

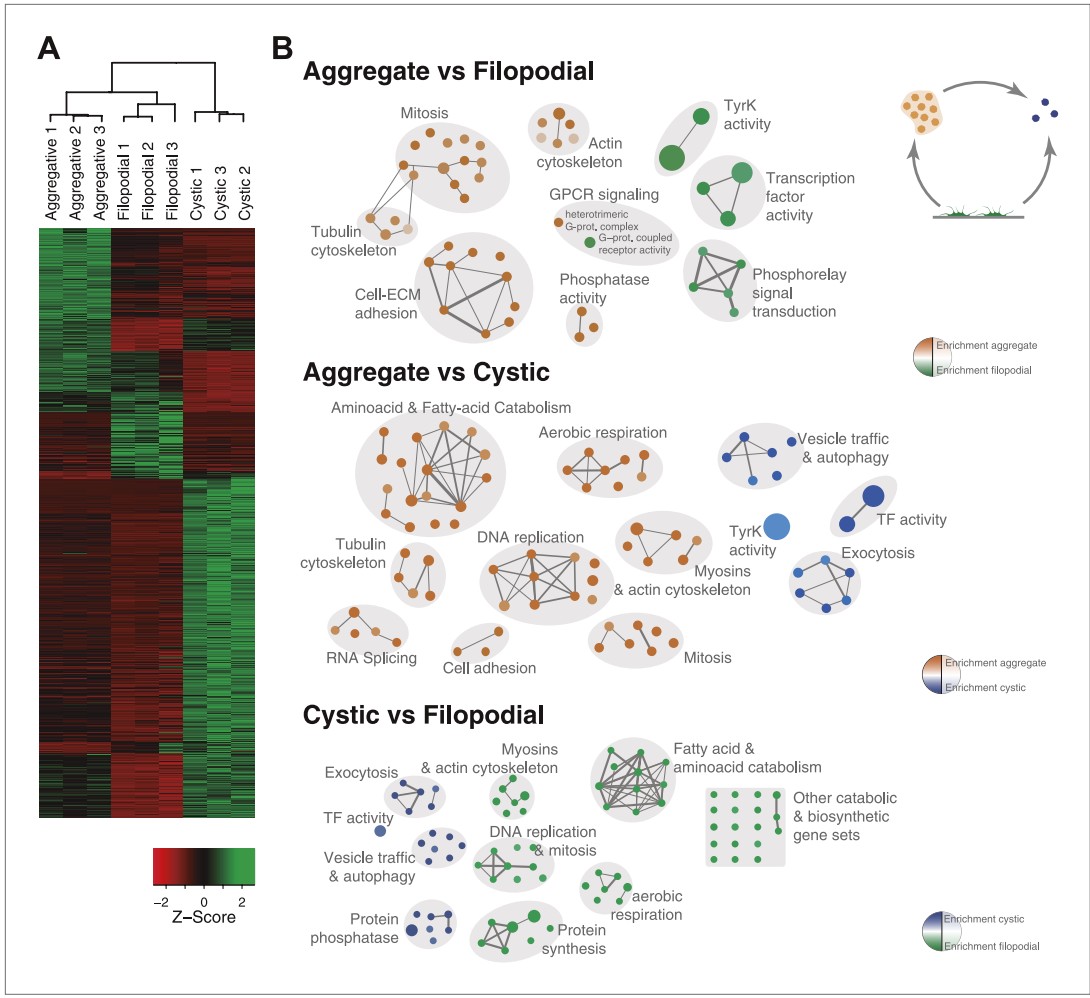

**Figure 5**. Differential gene expression in *C. owczarzaki*. (**A**) Heatmap showing differential gene expression in the different biological replicates of each stage. Only genes with cRPKM ≥ 5 in at least one sample and with a 2-fold expression change in at least one pair-wise comparison are shown. (**B**) Gene set enrichment analysis (GSEA) for the different cell stages ('Materials and methods'). Orange represents enrichment in the aggregative stage, blue in the cystic stage, and green in the filopodial stage, with color intensity proportional to enrichment significance. The node size is proportional to the number of genes associated to the GO category, and the width of the edges is proportional to the number of genes shared between GO categories. Groups of functionally related GOs are manually circled and assigned a label.

*C. owczarzaki* shows a highly-regulated expression pattern, with lowest expression in the filopodial stage (*Figure 10C*). Therefore, it is tempting to speculate that Elav-like protein may negatively regulate filopodial-specific intron retention of some introns. Experimental depletion of Elav-like protein in *C. owczarzaki* will require the development of RNAi or gene-targeting methods in this species before this hypothesis can be tested.

Next, we investigated differential exon splicing and identified 191 cassette exons with PSIs <95% in at least one life cycle stage, 39 of which display PSIs <85%. 29 of these exons showed a ≥15% PSI difference in pairwise comparisons between the cell stages, with lower PSIs typically associated with the filopodial stage (*Supplementary file 1*); RT-PCR assays confirmed skipping for 7 out of 8 tested cases (*Figure 11A*, and *Figure 11—figure supplement 1*). Most (~60%) of these exons maintain an open reading frame when skipped. In contrast to previous reports demonstrating that differentially-regulated exons are significantly under represented in modular, folded domains in metazoans (*Romero et al., 2006*; *Ellis et al., 2012*), two thirds of the differently regulated exons in *C. owczarzaki* overlap annotated domains (*Supplementary file 1*). Furthermore, genes with differential exon

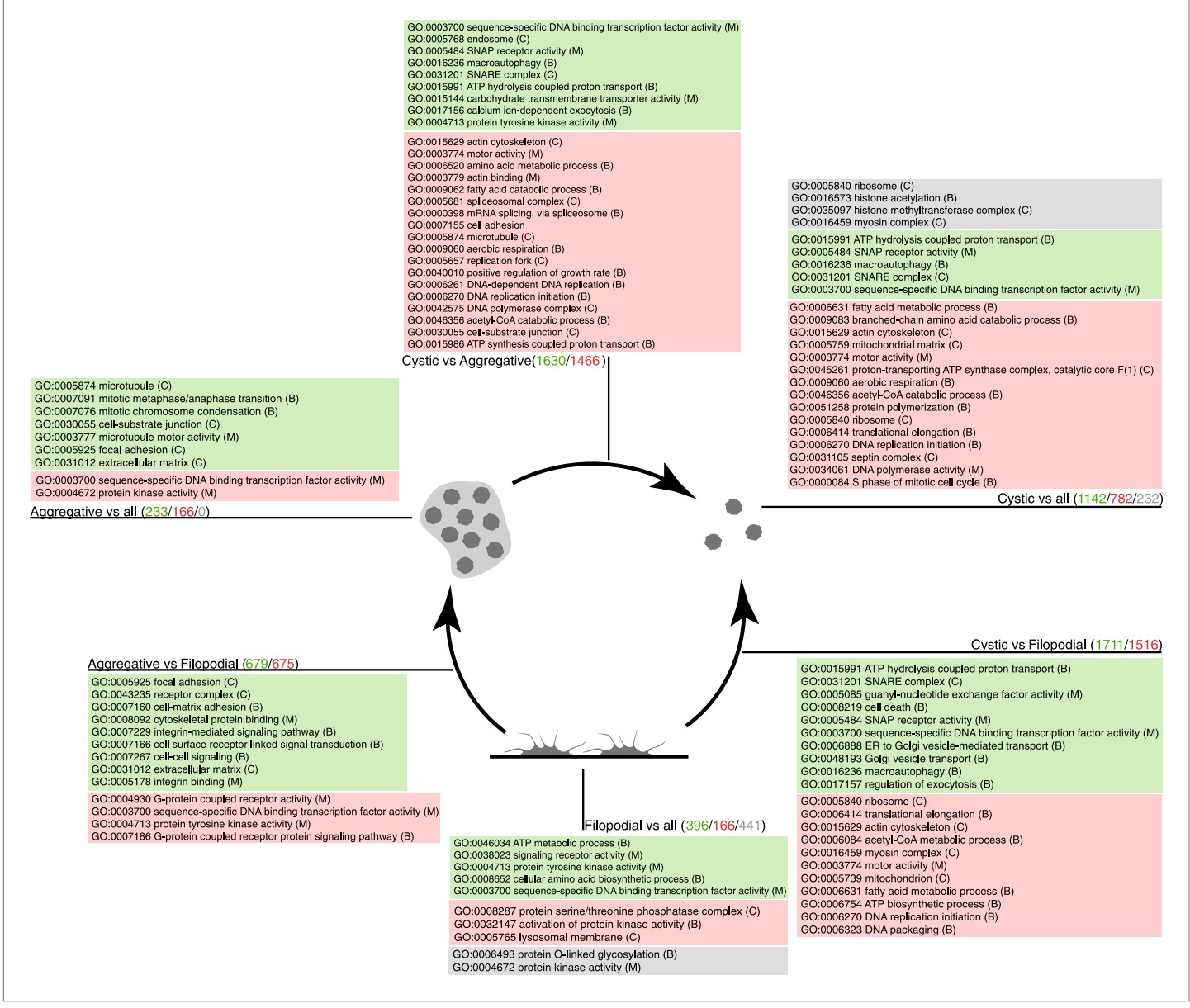

**Figure 6**. GO enrichment in sets of differentially expressed genes. Pairwise (Aggregative vs Filopodial, Cystic vs Aggregative and Cystic vs Filopodial) and one-versus-all comparisons are indicated. The significantly overrepresented GO categories ('Materials and methods') are shown for sets of overexpressed (green) and downregulated (red) genes and for genes with differential intron retention (gray). The number of genes included in each set is indicated with the same color code.

skipping are statistically significantly enriched in protein kinase activity, impacting both tyrosine and serine/threonine kinases (**Figure 11B**). This observation strongly suggests a role for coordinated exon skipping in the modulation of cell signaling in *C. owczarzaki*. To our knowledge, this represents the first example of a regulated exon network linked to a specific biological function in a unicellular organism.

In summary, our results offer new insight into the origin of metazoan multicellularity. In particular, the observation of an aggregative multicellular stage in *C. owczarzaki* represents the first example of such cellular behavior in a close unicellular relative of metazoans. This observation therefore adds to the repertoire of reported complex cellular behaviors among extant unicellular relatives of metazoans— including clonal colony formation in choanoflagellates and sporangia formation by hypertrophic syncytial growth in ichthyosporeans (**Dayel et al., 2011**; **Suga and Ruiz-Trillo, 2013**), thus expanding

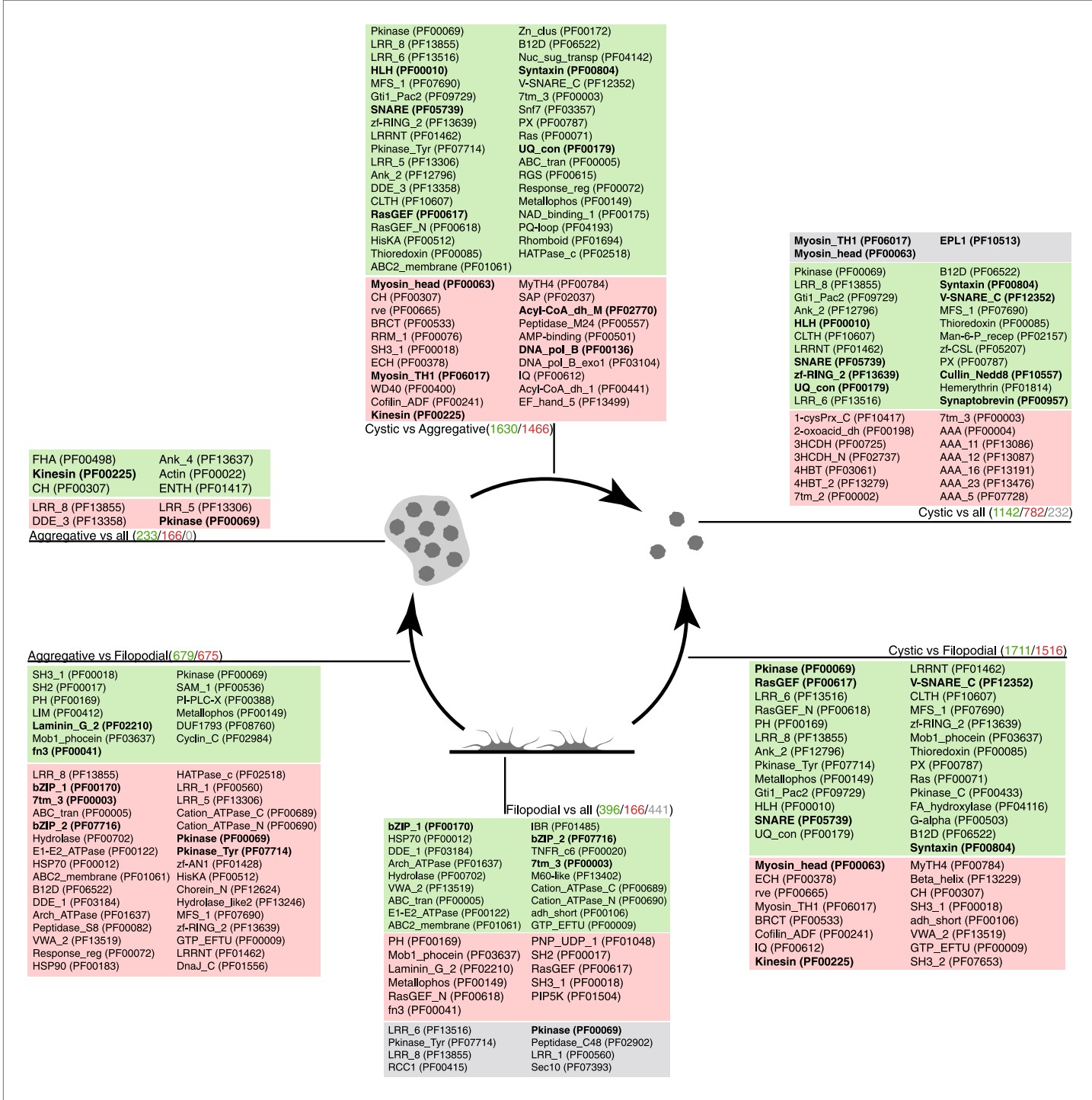

**Figure 7**. Pfam protein domain enrichment in sets of differentially expressed genes. Pairwise (Aggregative vs Filopodial, Cystic vs Aggregative and Cystic vs Filopodial) and one-versus-all comparisons are indicated. Significantly overrepresented Pfam domains ('Materials and methods') are shown for sets of overexpressed (green) and downregulated (red) genes and for genes with differential intron retention (gray). The number of genes included in each set is indicated with the same color code. Those Pfam domains mentioned in the text are shown in bold.

the potential starting 'raw material' available for the evolution of animal multicellularity. We note that the current evolutionary framework on the opisthokonts, based on phylogenomic analyses (*Steenkamp et al., 2006*; *Ruiz-Trillo et al., 2008*; *Shalchian-Tabrizi et al., 2008*; *Torruella et al., 2012*; *Paps et al., 2013*), discards the possibility that *C. owczarzaki* (or choanoflagellates) derives from a more complex

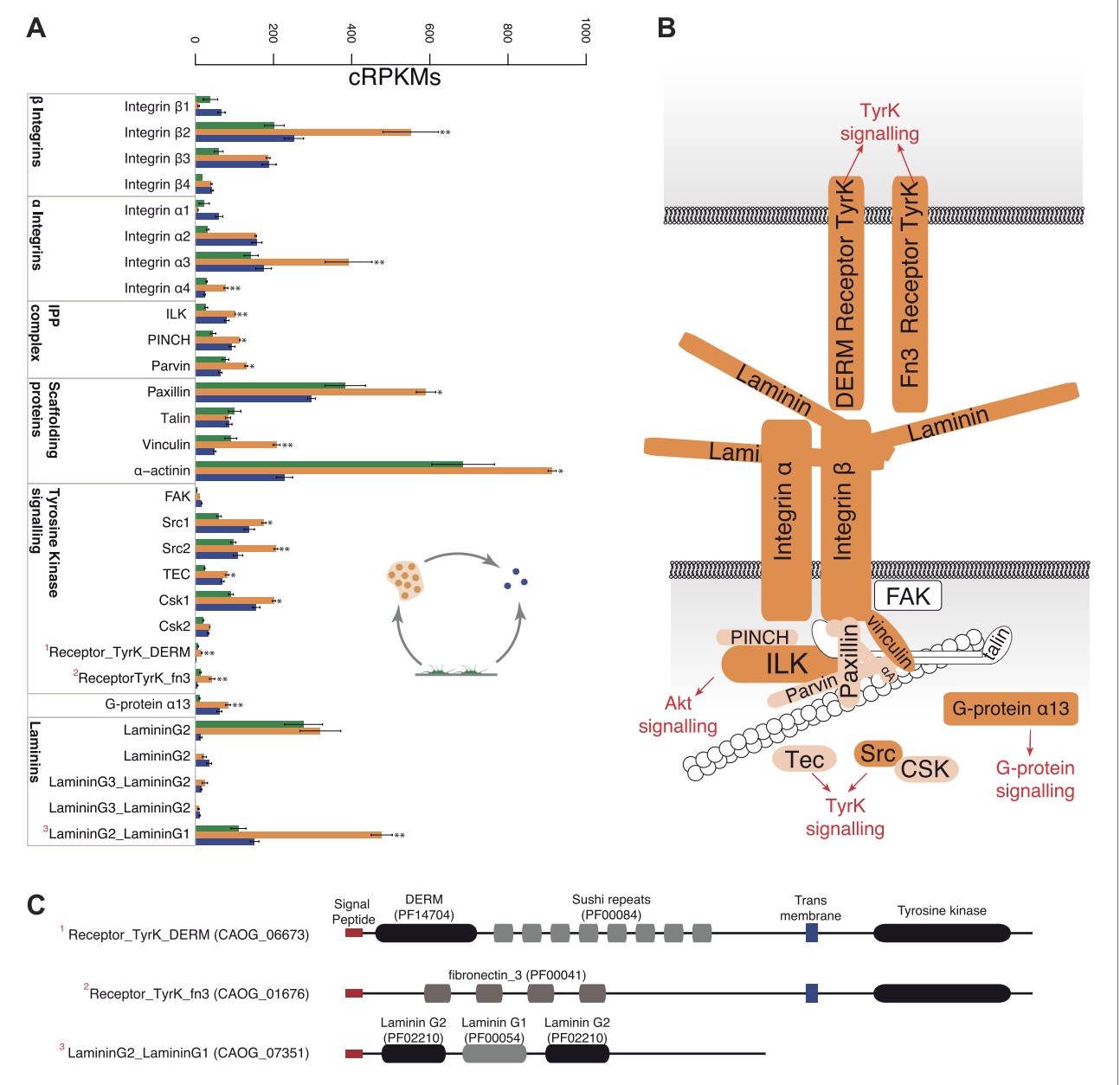

**Figure 8**. Expression of cell–ECM adhesion genes. (**A**) Barplot of the expression values of each gene in the different stages, showing overexpression of most components in the aggregate stage (orange). Asterisks indicate that the gene is significantly differentially expressed in both (two asterisks) or only one (one asterisk) pair-wise comparison (agg. vs fil. and agg. vs cyst.). Bars show standard error. (**B**) Schematic representation of the putative *C. owczarzaki* integrin adhesome and putative associated signalling proteins, colored according to overexpression in aggregates as shown in the barplot (dark orange, two asterisks; light orange, one asterisk; and white, no differences in expression). (**C**) Specific protein domain architectures for the fibronectin and DERM receptor tyrosine kinases (CAOG_01676 and CAOG_06673) and for the laminin protein (CAOG_07351).

multicellular lineage. The sister-group of opisthokonts is the unicellular biflagellates Apusozoa, and complex multicellularity has not yet been observed in any of the non-metazoan holozoan lineages.

Furthermore, we show that the complex, metazoan-like genetic 'toolkit' of *C. owczarzaki* (*Sebé-Pedrós et al., 2010, 2011*; *Suga et al., 2012*) is dynamically deployed during its highly-regulated life cycle, with upregulation of integrin adhesome and signalling genes linked to multicellularity in

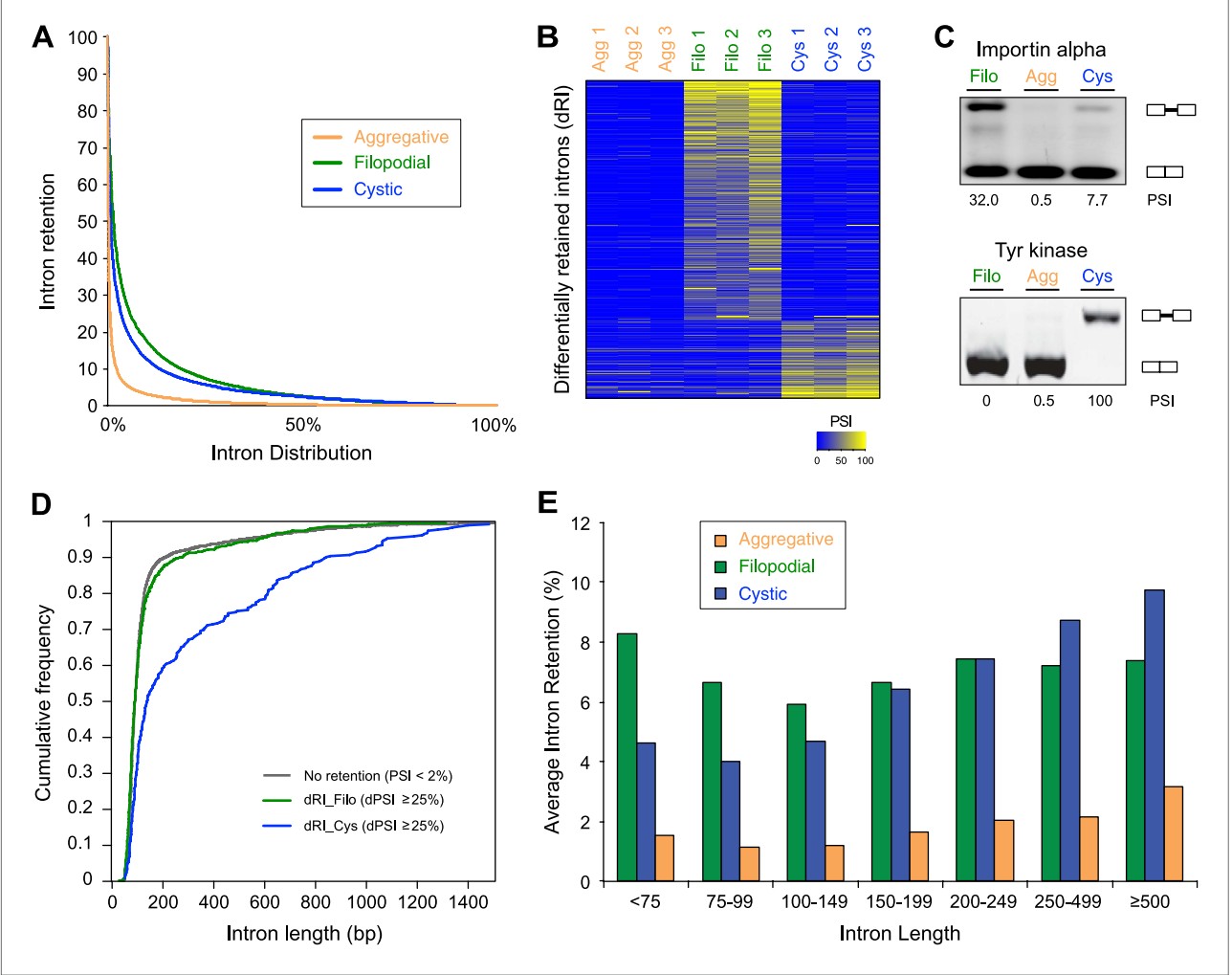

**Figure 9**. Regulated alternative splicing in *C. owzarzaki*. (**A**) Plot of percentage of intron inclusion by intron in rank order for the three studied cellular stages. Filopodial (green) and cystic (blue) stages show higher intron retention levels than the aggregative stage (orange) (p<2.2e-16, Wilcoxon Rank Sum test). (**B**) Heatmap of PSIs of filopodial- and cystic-specific differentially retained introns across three replicates for each cellular stage. (**C**) Examples of stage-specific intron retention. (**D**) Intron length distributions for differentially retained introns in cystic (blue), filopodial (green), and weakly retained introns (gray). (**E**) Relationship between intron length and retention. Percentage of average intron retention in each of the three cellular stages for different bins of intron size. In the cystic stage, the percentage of intron retention increased with intron length.

The following figure supplements are available for figure 9:

**Figure supplement 1**. Intron retention validation (see 'Materials and methods').

metazoans during the aggregative stage. Extensive differential AS between the *C. owczarzaki* life cycle stages likely further contributes to the dynamic gene regulation observed in this species, with differential intron retention likely acting as an important mechanism in the control of transcript levels between life cycle stages, probably through triggering non-sense mediated decay (NMD). Our discovery of an exon network associated with tyrosine kinase genes in *C. owczarzaki* further adds to the metazoan-like features of this species. Together with genes resembling those that function in metazoan multicellular processes, the emergence of an exon network that functions in conjunction with differentially-regulated intron retention may have provided a degree of proteomic and regulatory complexity that was key in the evolution of cell type complexity in metazoans (**Nilsen and Graveley, 2010**). Based on the collective results from our investigation of *C. owczarzaki*, it is intriguing to consider that the integration of regulatory innovations involving differential expression

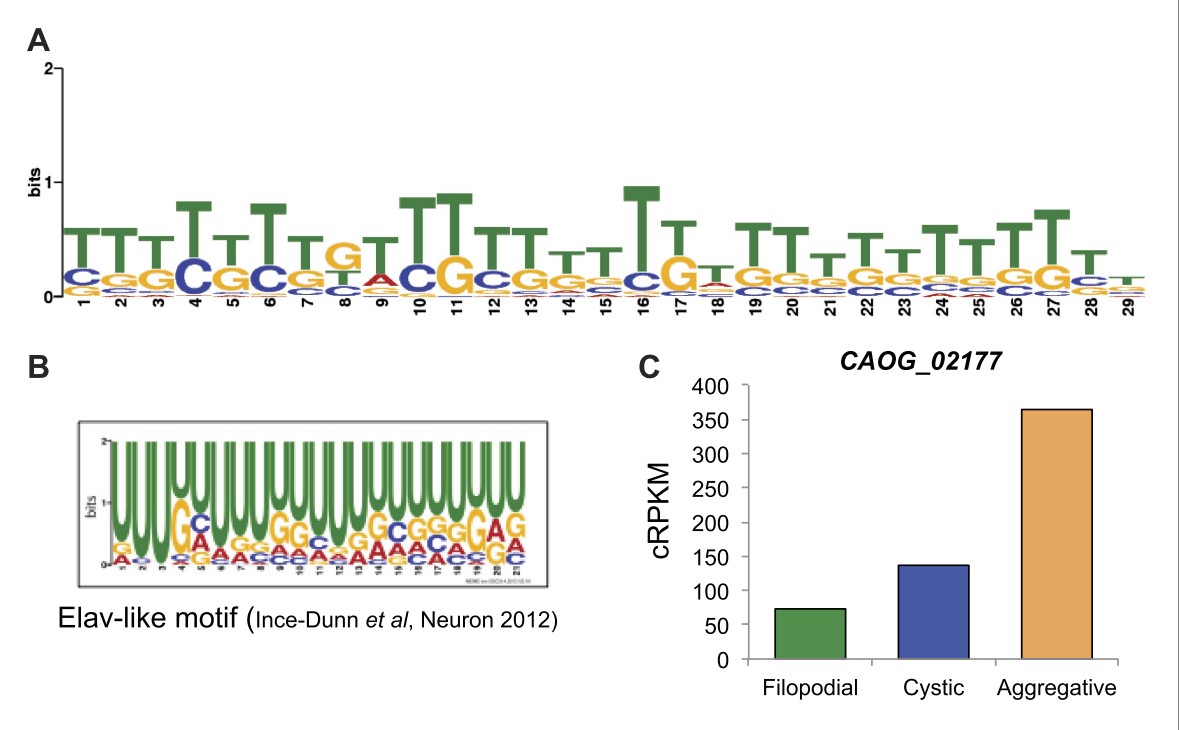

**Figure 10**. Possible role for an Elav-like ortholog in the negative regulation of filopodial stage-specific dRIs. (**A**) Most significantly enriched motif in filopodial stage-specific dRIs, obtained by MEME. (**B**) Consensus motif obtained by CLIP-Seq data for an Elav-like member in mammals by ***Ince-Dunn et al. (2012)*** and that closely resembles the motif in (**A**). T~U. (**C**) Expression (measured as cRPKMs) of CAOG_02177, a Elav-like ortholog from *C. owczarzaki* that shows lower expression in filopodial stage.

and splicing of metazoan-like genes set the stage for the evolution of cell specialization in the common ancestors of metazoans and *C. owczarzaki*.

## Materials and methods

### Scanning electron microscopy

*C. owczarzaki* cells of the corresponding stage were fixed for 1 hr with 2.5% glutaraldehyde (Sigma-Aldrich, St. Louis, MO, USA), and for another hour with 1% osmium tetroxide (Sigma-Aldrich), followed by dehydration in a graded ethanol series (25%, 50%, 70%, 99%) for 15 min per step, followed by three 15-min rinses in 100% ethanol. Samples were critical-point dried in liquid $CO_2$ using a BAL-TEC CPD 030 critical-point drying apparatus. They were subsequently glued to SEM stubs with colloidal silver, sputter-coated with gold-palladium, and examined with a Hitachi S-3500N (Hitachi High-Technologies Europe GmbH, Krefeld, Germany).

### Transmission electron microscopy

Cell aggregates were loaded into the copper tubes and immediately cryoimmobilized using a Self-Pressurized Freezing System (EM SPF) (Leica-Microsystems, Vienna, Austria). The cells were then stored in liquid nitrogen until further use. Peeled copper tubes were freeze-substituted in anhydrous acetone containing 2% osmium tetroxide and 0.1% uranyl acetate at −90°C for 72 hr and warmed to room temperature, following a 2°C increase per hour in five consecutive steps (−60°C, −30°C, 0°C, 4°C, and room temperature) with a total of 8 hr at each temperature and using an EM AFS (Leica-Microsystems, Vienna). After several acetone rinses, samples were infiltrated with Epon resin during 7 days and embedded in resin and polymerised at 60°C during 48 hr. Ultrathin sections were obtained using a Leica Ultracut UC6 ultramicrotome (Leica-Microsystems) and mounting on Formvar-coated copper grids. The sections were stained with 2% uranyl acetate in water and lead citrate, and were observed in a Tecnai Spirit 120 kv electron microscope (FEI Company, Eindhoven, Netherlands) equipped with a Megaview III CCD camera.

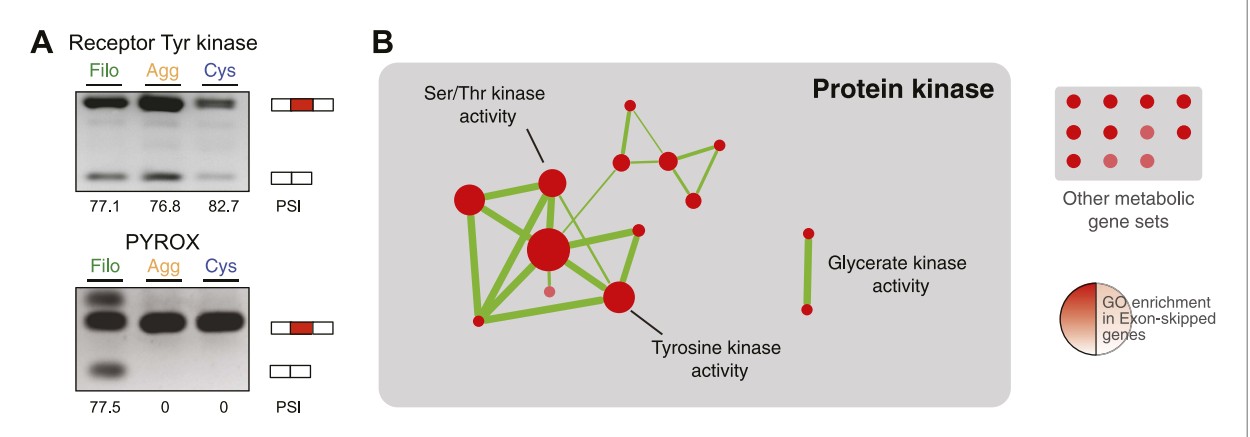

**Figure 11**. Regulated exon-skipping in *C. owzarzaki*. (**A**) Examples of exon skipping. (**B**) Gene set enrichment analysis (GSEA) of the genes containing cassette exons that are differentially-regulated among cellular stages showing high enrichment for protein kinase-associated activities.

The following figure supplements are available for figure 11:

**Figure supplement 1**. Exon Skipping validation by RT-PCR ('Materials and methods').

## Cell culture conditions

*C. owczarzaki* cells were grown axenically in 5-ml flasks with ATCC medium 1034 (modified PYNFH medium) in a 23°C incubator. Three biological replicates (three independent cell lines) were generated by subculturing from a single-founding cell and grown for 2 months. Adherent filopodiated cells were obtained by initiating a new 1/100 sub-culture (from an approximately $5 \times 10^6$ cells/ml initial culture) and, after 3–4 days, cells were scratched from the substrate. Aggregate formation was induced by initiating a new 1/250 sub-culture (from an approximately $5 \times 10^6$ cells/ml initial culture) and by gentle agitation at 60 rpm during 4–5 days. Finally, floating cystic cells were obtained from a 14-day-old culture, starting from a new 1/100 sub-culture (from an approximately $5 \times 10^6$ cells/ml initial culture).

## Aggregation experiments

Two different groups of cells (from two different starting cultures, 5 ml flasks with a cell density of $10^6$ cells/ml, consisting exclusively of adherent filopodial cells) were stained either with 75 nM (in PBS1x) Lysotracker Green DND-26 (Life Technologies, Carlsbad, CA, USA) or with 1 µM (in PBS1x) Chromeo Live Cell Mitochondrial Staining Kit (Active Motif Inc, Carlsbad, CA, USA). The cells were stained for 30 min at 23°C. After staining, 1/3 of the cells from the two differentially stained populations were mixed in a new culture flask and the remaining 2/3 of cells for each staining were kept as a control. All three cultures were grown for 2 hr and then the aggregate formation was induced (see above, 'Results and Discussion'). After 8 hr, aggregates were visualized in poly-L-lysine covered (Sigma-Aldrich, St. Louis, MO) glass-bottom plates in a Leica TCS SP5 confocal microscope (Leica-Microsystems).

C. owczarzaki cell division was blocked using 100 mM hydroxyurea (Sigma-Aldrich) or 25 µg/ml aphidicolin (Sigma-Aldrich). The effect of both drugs was evaluated by following cultures treated with each drug during 7 days, using Neubauer chamber. The cells were cultured in 16-multiwell plates and with an initial density of $5 \times 10^4$ cells/ml. Once these conditions were established, different cultures were treated with each drug for 1 day and, then, aggregate formation was induced (see above). 2 days later, the formation of aggregates was visualized in poly-L-lysine covered (Sigma-Aldrich) glass-bottom plates in a Leica TCS SP5 confocal microscope (Leica-Microsystems).

## RNA-Seq and analysis

*C. owczarzaki* cells were grown in 5-ml flasks with ATCC medium 1034 (modified PYNFH medium) in a 23°C incubator. Total RNA from each cell stage (and from three biological replicates from each stage) was extracted using Trizol reagent (Life Technologies). Nine libraries were sequenced over 2 lanes HiSeq 2000 instrument (Illumina, San Diego, CA, USA), generating a total of 197M 76-base paired

reads. Reads were aligned to the reference genome using Tophat (*Trapnell et al., 2012*) with default options and specifying that this was a strand-specific sequencing, rendering an average mapping of 90%. Significant differential expression was calculated by performing pairwise comparisons with DESeq (*Anders and Huber, 2010*) (threshold 1e-05), EdgeR (*Robinson et al., 2010*) (threshold 1e-05), CuffDiff (*Trapnell et al., 2012*) (threshold 1e-05) and NOISeq (*Tarazona et al., 2012*) (threshold 0.8) and only genes that appear to be significant at least in three out of the four methods were considered as differentially expressed. Quality control analyses of the data were performed using cummeRbund R package (*Trapnell et al., 2012*). These include count vs dispersion plot to estimate over-dispersion, density plot to assess the distributions of FPKM scores across samples and squared coefficient of variation plot to check for cross-replicate variability.

A gene ontology of *C. owczarzaki*'s 8637 genes was generated using Blast2GO (*Conesa et al., 2005*) and GO enrichments of the different lists of differentially expressed genes (see above) were analyzed using Ontologizer (*Bauer et al., 2008*) using the Topology-Weighted method. A p-value threshold of 0.01 was used. The results from Ontologizer were loaded into Enrichment map cytoscape plug-in (*Merico et al., 2010*) to generate a network visualization. Pfam domains of all genes were analyzed using Pfamscan v26 with default Gathering Threshold, and counts were generated using custom Perl scripts. Fisher's exact tests were performed using custom R scripts and a p-value threshold of 0.01 was used.

## Alternative splicing analysis

Exon skipping and intron retention were analyzed as previously described (*Curtis et al., 2012*; *Han et al., 2013*). In short, for exon-skipping analyses, multifasta libraries of exon–exon junctions were built by combining all forward annotated splicing donors and acceptors. A minimum of eight base pairs was required at each boundary to assure specificity. Next, the number of effective mappable positions was calculated for each exon–exon junction, as previously described (*Labbé et al., 2012*; *Barbosa-Morais et al., 2012*). Then, RNA-seq reads (previously trimmed to 50 nucleotides and combining each three replicates to increase read depth) were aligned to these sequences using Bowtie, with −m 1 −v 2 parameters (single mapping and two or fewer mismatches). Percentage of exon inclusion was calculated and a minimal read coverage was required, as previously described (*Khare et al., 2012*). For intron retention, a similar approach was taken for each contiguous intron–exon and exon–exon junction, and percentage of intron inclusion (PSI, Percent Spliced In, the percentage of transcript for a given gene that contain the intron) was calculated as previously described (*Curtis et al., 2012*). For comparisons among cellular stages, only events with enough read coverage in the three samples were considered (either (i) ≥10 reads in the exon–exon junction or (ii) ≥10 reads in one intron–exon junction and ≥5 in the other), and introns showing >95% inclusion in the three samples were discarded. To assess whether differentially retained introns in the same genes were included in a coordinated or in a combinatorial manner, mate information of read pairs was used. If each end of a read mapped to two different intron retention events, each end may be providing support for retention of both introns or splicing of both introns (coordinated regulation), or retention of one and splicing of another (combinatorial intron retention). For the 555 pairs of retained introns that had read mate information, 196 (35.3%) showed evidence for combinatorial regulation. Finally, for sequence motif enrichment analyses, full intron sequences were compared using MEME (*Bailey et al., 2009*).

## RT-PCR

To validate AS analysis predictions, the three stages were induced (RNA-Seq and analysis sections) and RNA was extracted using Trizol reagent (Life Technologies). To eliminate genomic DNA, total RNA was treated with DNAse I (Roche, Basel, Switzerland) and purified using RNeasy columns (Qiagen, Venlo, Netherlands). For each stage, cDNA was produced from 1 µg of total RNA using SuperScript III reverse transcriptase (Life Technologies). Pairs of primers of similar melting temperature (60°C) and spanning the putative alternatively spliced segments were designed using Geneious software. PCR was performed using ExpandTaq polymerase (Roche).

## Flow cytometry

*C. owczarzaki* cells were grown for 10 to 15 days, sampling every day from both the supernatant (to obtain floating cells, which after day 7 are completely encysted) and the scratched flask (to obtain filopodial adherent cells). Thus, two samples were obtained daily, for floating and adherent cells. For DNA-content analysis, a sample was fixed using 70% ethanol and stored at −20°C for one month. The

samples were subsequently fixed and stained with Propidium Iodide (as described in *Darzynkiewicz and Huang, 2004*) and DNA content estimated using FACScalibur flow cytometer (Becton Dickinson, Franklin Lakes, NJ, USA). For cell counting, 1 ml of fresh sample (one from the supernatant and one from the flask surface) was mixed in a BD Trucount Tube (Becton Dickinson), with a known number of beads, so absolute cell number counts could be calculated, using an LSR Fortessa flow cytometer (Becton Dickinson). Two replicate experiments (R1 and R2) were performed independently in order to confidently establish growth dynamics. Two measures were calculated from the DNA-content analysis. First, the proliferation rate, which indicates the proportion of number of cells in S and G2/M phases vs the number of cells in G0/G1. Second, the percentage of cells in S-phase.

## Acknowledgements

We thank Joshua Levin and Lin Fan for generating RNA-Seq libraries and the Broad Institute Genomics Platform for Illumina sequencing. The authors are grateful to Carmen López Iglesias of the Cryo-Electron Microscopy unit (CCiT-UB) for her help and advice on electron microscopy. We also thank Cristina Peligero, for her help in performing and interpreting the flow cytometry results, Ignacio Maeso for critical reading of the manuscript, and Professor Yves van de Peer, Stephane Rombauts and Brian Haas for their advice. ASP is supported by a pregraduate Formacion Profesorado Universitario grant from MICINN. MI is supported by a postdoctoral fellowship from the Human Frontiers Science Program Organization. BJB acknowledges grant funding from the Canadian Institutes of Health Research.

## Additional information

### Competing interests

BJB: Reviewing editor, *eLife*. The other authors declare that no competing interests exist.

### Funding

| Funder | Grant reference number | Author |
| --- | --- | --- |
| European Commission | ERC-2007-StG- 206883 | Iñaki Ruiz-Trillo |
| Ministerio de Economia y Competitividad | BFU2011-23434 | Iñaki Ruiz-Trillo |
| Canadian Institutes of Health Research | MOP-67011 | Benjamin J Blencowe |

The funders had no role in study design, data collection and interpretation, or the decision to submit the work for publication.

### Author contributions

AS-P, MI, Conception and design, Acquisition of data, Analysis and interpretation of data, Drafting or revising the article; JC, HP-A, Acquisition of data, Analysis and interpretation of data, Drafting or revising the article; CR, Acquisition of data, Drafting or revising the article; CN, Conception and design, Acquisition of data, Drafting or revising the article; BJB, IR-T, Conception and design, Analysis and interpretation of data, Drafting or revising the article

## Additional files

### Supplementary files

• Supplementary file 1. Exon-skipping events in *C. owczarzaki*. For each event, the table shows the affected gene, a reference of the skipping event, the coordinates of the skipped exon, its length and if it is multiple of three nucleotides, the full coordinates of the event (i.e., including the donor of the upstream constitutive exons and acceptor of the downstream constitutive exon), percentage of inclusion of the exon among all transcripts present at each stage, general protein domain structure and a brief description of the gene, and information on which region of the protein is affected by the exon-skipping event (in orange if impacting a structured domain). Protein kinases are highlighted in dark green and tyrosine kinases in light green.

## Major dataset

The following dataset was generated:

| Author(s) | Year | Dataset title | Dataset ID and/or URL | Database, license, and accessibility information |
| --- | --- | --- | --- | --- |
| Russ C, Nusbaum C, Sebé-Pedrós A | 2013 | RNAseq data from *Capsaspora owczarzaki* | http://www.ncbi.nlm.nih.gov/biosample/?term=txid595528[Organism:noexp] | Publicly available at NCBI (http://www.ncbi.nlm.nih.gov/) |

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
