## [Decision Letter]

Thank you for sending your work entitled “Regulated multicellularity in a close unicellular relative of Metazoa” for consideration at *eLife*. Your article has been favorably evaluated by a Senior editor and 3 reviewers, one of whom is a member of our Board of Reviewing Editors.

The Reviewing editor and the other reviewers discussed their comments before we reached this decision, and the Reviewing editor has assembled the following comments to help you prepare a revised submission:

The paper describes a new aggregative stage for a unicelllular taxon (*Capsospora*). Because of its relationship to animals and the Choanoflagellata, *Capsaspora* is becoming an increasingly important model organism for reconstructing animal origins. For example, *Capsaspora* has a number of genes in its genome that seem to be missing from the Choanoflagellata. What has not been clear is the extent to which the study of *Capsaspora* cell biology (beyond the study of its genome) will help to illuminate animal origins.

This paper begins to address that question by describing the life history of *Capsaspora*, with a focus on its ability to form multicellular aggregates. The images in Figure 1 nicely document the morphology of various differentiated cell types. The authors further report that the life history of *Capsaspora* is associated with regulated and coordinated changes in transcriptional regulation and in mRNA splicing. The transition to cell aggregation in particular is associated with differential transcription of genes that are also involved in the regulation of development and cell adhesion in animals, raising the possibility that the clonal multicellularity of animals evolved through the co-option of genes that previously functioned in cell aggregation.

However, the manuscript in its present form does not entirely document several of the key bioinformatics analyses, such that it is difficult to assess if their results are truly meaningful, or if there is cherry picking in the results. A primary concern is that the authors observe that a significant portion of the genome is differentially transcribed during the life cycle of *C. owczarzaki*, yet they are making the leap to very specific molecular pathways that are linked to these life cycle stages. As the analyses to generate gene expression lists that are significantly correlated are not fully documented, it is impossible to evaluate if the authors are indeed making a significant advance in our understanding of multicellular evolution, or if they are selectively highlighting one of many interesting pathways that change during the life cycle of *C. owczarzaki*.

A common theme in the comments of the referees concerns a lack of adequate documentation for important assertions in the paper. Some of these inadequately supported assertions are:

1) that colonies form through aggregation

2) that colonies form in a density-dependent manner and are homogeneous

3) that the gene families and networks they highlight are important to the regulation of life history transitions

4) that multicellularity in *Capsaspora* has the potential to be informative about the origin of multicellularity in animals.

The following substantial concerns were raised by referees #2 and #3 and need to be addressed in a revised version. Please pay particular attention to the point that “Lists of genes” are not particularly useful on their own; what is key are the functions associated with evolving multicellularity. The referees feel that this is the crucial aspect that should come out of this paper to make it suitable for publication in *eLife*.

*Reviewer #2*:

1) The authors assert that the multicellular stage of the *Capsaspora* life cycle forms through aggregation, but the evidence for it (Videos 2 and 3) is unconvincing. I see that clumps are forming, but it is possible that the clumps could be forming clonally (i.e., through cell division) rather than through aggregation. A more convincing presentation would be higher magnification time-lapse microscopy and false-coloring of individual cells prior to aggregation to show that they transition from being solitary to being members of an aggregant. Even this would only show that aggregation is possible, while not eliminating the possibility that multicellular forms of *Capsaspora* can also develop through cell division.

2) It would be helpful if the authors would devote more space to discussing plausible alternative scenarios regarding the origin of multicellularity along the animal stem lineage. The authors imply that the last common ancestor of animals and *Capsaspora* had aggregative behavior (e.g., see the last sentence of the Abstract), but it is also possible that aggregation in *Capsaspora* is derived within the Filasterea. The argument in favor of an “aggregative” ancestor would be strengthened by citing and discussing the work of James Nelson's group on *Dictyostelium* (e.g. ,PMID 22930590 and PMID 21393547). On the other hand, the study of *Creolimax* from the corresponding author's lab shows another, alternative route to multicellularity, in which a syncytium forms and is subsequently cellularized. In this case, the multicellular form is clonal, as it is in Choanoflagellata and animals. Because *Creolimax* is an outgroup of the clade containing *Capsaspora*, Choanoflagellata, and animals, it is plausible that their last common ancestor formed multicellular structures clonally and that the aggregative process saw *Capsaspora* evolved within the Filasterea. Some of these alternative scenarios were nicely discussed in the *Creolimax* paper and should be revisited in this paper in light of the results from *Capsaspora*.

3) The transcriptional regulation/splicing section of the paper. This comes down to a matter of taste, and the state of the field of functional genomics. I found this section to be useful as a platform for future research (it is obviously good to know which genes are turned on and off or alternatively spliced, and when), but not particularly insightful. For example, it's not clear which specific genes regulate aggregation, nor is it clear whether aggregation in *Capsaspora* is homologous with the aggregative behaviors seen in some animal cells. In past functional genomics papers it was considered interesting to know which GO terms were enriched, but I think these types of analyses are now better suited to specialist journals. The data and findings from the RNA-seq experiments should definitely be published, but I'm not sure that they are of sufficient general interest as currently presented. A better distillation of the data might reveal a gem that is currently buried. Perhaps a more clear presentation and emphasis of the alternative splicing data would help.

*Reviewer #3*:

1) A key limitation of this work is that the authors show that approximately half the genome is differentially transcribed at each of the life cycle stages. The authors then somehow whittle these genes down to a list of candidate genes that they associate as being important for each of the life cycle stages. While these analyses seem to draw interesting conclusions, it is not apparent that the authors have actually tested against a null hypothesis.

2) Functional testing of these candidate genes has not been done (e.g., RNAi of the candidate genes to test for life cycle defects). However, it may not be possible in this non-model system. If functional analyses cannot be done then it is critical that the authors carefully analyze and document their methods so they can move beyond “lists of genes” and toward the functional insights they have made. This major criticism goes hand-in-hand with point #4 that the bioinformatics analyses need more explanation. I would be less worried about this point if point #3 were to be comprehensively addressed in a revised manuscript.

3) [Supplementary-material SD1-data]: CAOG_03572, is the isoform above the exon skipped form? CAOG_07148, what is the isoform in the Filo stage just above the exon skipped isoform? CAOG_05784, What are the isoforms just below the inclusion isoform? Either the RT-PCR is not working as intended or there are potentially many other splice isoforms present.

4) There are several instances where the authors cite “significant enrichment of” an annotated function where they point to network diagrams presumably derived from cytoscape. As these are key conclusions in the paper, documentation of how these were derived as cytoscape analysis of pathway enrichment is not a standardized analysis. Data specifically noted are Figure 4 and Figure 9. My concern with these types of enrichment analyses are if the authors are specifically looking at genes they are interested in, or are annotated, versus actually testing against the null hypothesis. For example, are protein kinases actually more likely to have exon skipping than just any random set of genes in the genome (Blast2Go annotated or not)? What is the actual meaning of “genes with differential exon skipping are significantly enriched in protein kinase activity” and other related statements? These are key conclusions to this paper and as submitted it is difficult to evaluate if these statements are indeed true. Indeed, the authors claim, regarding protein kinases, that “[t]his represents the first example of a regulated exon network linked to a specific biological function in a unicellular organism”. This conclusion needs more documentation of how it was reached for me to agree.

---

## [Author Response]

*A common theme in the comments of the referees concerns a lack of adequate documentation for important assertions in the paper. Some of these inadequately supported assertions are*:

*1) that colonies form*
*through aggregation*

A major argument that colonies form by aggregation rather than clonal division is that the aggregates are detected within a few hours, whereas the doubling time of a normal culture during exponential growth is approximately 24 hours. This is shown in Video 1. To provide further support for the formation of aggregates we have provided results from additional experiments. First, we used two live staining protocols (Chromeo Live Cell Mitochondrial Staining Kit and Lysotracker Green DND-26) that appear to be efficient (staining virtually all cells in the culture), stable (remaining for days), and non-toxic (with the cells exhibiting normal morphological shape and behavior) in *C. owczarzaki*. We mixed two populations of differentially stained cells (one with Chromeo Live Cell and another with Lysotracker Green) and obtained aggregates with cells from both populations, indicating that colonies form indeed by aggregation of multiple individuals. This experiment is described in the text and shown in the new Figure 4.

Second, we performed DNA content analyses by flow cytometry of aggregates and found that the proliferation rate is negligible, indicating that little cell division occurs in aggregates. This result is also now included in Figure 4—figure supplement 1.

Finally, we show that the cell cycle blocking drugs hydroxyurea and aphidicolin that effectively block cell division in *C. owczarzaki* do not prevent formation of cell aggregates. This strongly indicates that clonal division is not required for the formation of aggregates. These new results are included in the new Figure 4.

Given the strength of our new evidence of aggregative multicellularity, we have modified the title of the manuscript as follows:

“Regulated aggregative multicellularity in a close unicellular relative of Metazoa”.

*2) that colonies form in a density-dependent*
*manner and are homogeneous*

We agree that the evidence to support this claim is not strong and therefore have removed the statement from the manuscript. It should also be noted that this is not a major point of our paper; addressing it would be very time consuming yet any outcome would not alter the main conclusions of our manuscript.

*3) that the gene families and networks*
*they highlight are important to the regulation of life history transitions*

There are, unfortunately, no available genetic tools in *Capsaspora* to specifically address this question at a molecular level. However, we believe that this does not undermine our main novel message, which is to highlight that transitions between the different cell stages of *Capsaspora* are highly regulated at the transcriptomic level and that this regulation has features in common with metazoans.

We have, nonetheless, made a strong effort to make the datasets as comprehensive and informative as possible to serve as a basis for future research, as suggested by Reviewer 2.

*4) that multicellularity in* Capsaspora *has the potential to be informative about the origin of multicellularity in animals*.

We may have failed to stress this in the previous version and have thus modified several parts of the manuscript to address this point. We strongly believe that studying different close unicellular relatives of animals (rather than just one) is the only way to infer ancestral cellular and genomic characteristics of the unicellular ancestor of metazoans. Therefore, our new description of unique features in the life cycle of *Capsaspora* further expands the possible range of cell types and behaviors (the “starting raw material”) of the unicellular ancestor of Metazoa. Moreover, the high regulation that we observe and describe (which affects metazoan-like genes), provides a new framework with which to further analyze the role that gene regulation may have played in the unicellular to multicellular transition that gave rise to metazoans. We have expanded the Discussion with a new paragraph to address the reviewer’s point.

Reviewer #2:

*1) The authors assert that the multicellular stage of the* Capsaspora *life cycle forms through aggregation, but the evidence for it (*Videos 2 and 3*) is unconvincing. I see that clumps are forming, but it is possible that the clumps could be forming clonally (i.e., through cell division) rather than through aggregation. A more convincing presentation would be higher magnification time-lapse microscopy and false-coloring of individual cells prior to aggregation to show that they transition from being solitary to being members of an aggregant. Even this would only show that aggregation is possible, while not eliminating the possibility that multicellular forms of* Capsaspora *can also develop through cell division*.

Please see response to Point 1 above.

*2) It would be helpful if the authors would devote more space to discussing plausible alternative scenarios regarding the origin of multicellularity along the animal stem lineage. The authors imply that the last common ancestor of animals and* Capsaspora *had aggregative behavior (e.g., see the last sentence of the Abstract), but it is also possible that aggregation in Capsaspora is derived within the Filasterea. The argument in favor of an “aggregative” ancestor would be strengthened by citing and discussing the work of James Nelson's group on* Dictyostelium *(e.g., PMID 22930590 and PMID 21393547). On the other hand, the study of* Creolimax *from the corresponding author's lab shows another, alternative route to multicellularity, in which a syncytium forms and is subsequently cellularized. In this case, the multicellular form is clonal, as it is in Choanoflagellata and animals. Because* Creolimax *is an outgroup of the clade containing* Capsaspora*, Choanoflagellata, and animals, it is plausible that their last common ancestor formed multicellular structures clonally and that the aggregative process saw* Capsaspora *evolved within the Filasterea. Some of these alternative scenarios were nicely discussed in the Creolimax paper and should be revisited in this paper in light of the results from* Capsaspora.

We agree with the reviewer that this is an interesting point of discussion and speculation, and we have now extended our discussion of this topic at the end of the first paragraph of the Results and Discussion.

*3) The transcriptional regulation/splicing section of the paper. This comes down to a matter of taste, and the state of the field of functional genomics. I found this section to be useful as a platform for future research (it is obviously good to know which genes are turned on and off or alternatively spliced, and when), but not particularly insightful. For example, it's not clear which specific genes regulate aggregation, nor is it clear whether aggregation in Capsaspora is homologous with the aggregative behaviors seen in some animal cells. In past functional genomics papers it was considered interesting to know which GO terms were enriched, but I think these types of analyses are now better suited to specialist journals. The data and findings from the RNA-seq experiments should definitely be published, but I'm not sure that they are of sufficient general interest as currently presented. A better distillation of the data might reveal a gem that is currently buried. Perhaps a more clear presentation and emphasis of the alternative splicing data would help*.

The referee is right to point out that there is no clear homology between the aggregative behavior seen in *Capsaspora* and the one seen in some animal cells, yet we respectfully disagree that our data on the transcriptiomic regulation of *Capsaspora*’s aggregative behavior is not insightful. A major open question raised by the finding that *Capsaspora* has a complex repertoire of genes involved in multicellularity (Suga et al. Nat Commun 4, 2325 (2013)), as also seen for choanoflagellates (King et al. Nature 451, 783-788 (2008); Fairclough et al. Genome Biol 14, R15 (2013)) is whether differential gene regulation played an important role in the origins of multicellularity; however, gene regulation among unicellular relatives of Metazoa remains poorly understood. In our manuscript we provide the first description of transcriptomic changes associated with transitions between different cell stages in *Capsaspora*, both at the level of mRNA expression and alternative splicing. The nature of the genes that are differentially regulated is highly suggestive of possible mechanisms that underlie origins of multicellular. As such, we strongly feel that our observations provide an important basis for future investigation into animal origins. We have made changes to the text (as summarized above) to more clearly present our findings and their importance in terms of evolutionary implications.

Reviewer #3:

*1) A key limitation of this work is that the authors show that approximately half the genome is differentially transcribed at each of the life cycle stages. The authors then somehow whittle these genes down to a list of candidate genes that they associate as being important for each of the life cycle stages. While these analyses seem to draw interesting conclusions, it is not apparent that the authors have actually tested against a null hypothesis*.

We agree that this was not clear enough. We want to stress that we did not follow a “candidate”-driven approach, as the reviewer suggests. Instead, we used statistical enrichment tests (four of them) for both Gene Ontology terms and also all Pfam domains. The results of these tests were manually curated only to remove redundancy (in the case of nested GO terms). Therefore, the enriched functional categories that we report were found in a completely unbiased way. Then, we selected some categories that were significantly enriched in one stage and that were particularly interesting from an evolutionary perspective, such as the integrin adhesome, and examined them more closely gene-by-gene. We have modified the Materials and methods to clarify how we performed our analyses.

*2) Functional testing of these candidate genes has not been done (e.g., RNAi of the candidate genes to test for life cycle defects). However, it may not be possible in this non-model system. If functional analyses cannot be done then it is critical that the authors carefully analyze and document their methods so they can move beyond “lists of genes” and toward the functional insights they have made. This major criticism goes hand-in-hand with point #4 that the bioinformatics analyses need more explanation. I would be less worried about this point, if point #3 were to be comprehensively addressed in a revised manuscript*.

We completely agree with the reviewer that functional testing such as knocking- down some of the candidate genes that regulate life cycle transitions would be ideal. However, and as the reviewer rightly guesses, no such methods are yet available for *Capsaspora*, and addressing this would be beyond the scope of the present manuscript, given the major technical challenges involved. To address the reviewer’s other concern, we have provided a more detailed description of the analysis and statistical methods used in our paper.

*3)*
[Supplementary-material SD1-data]*: CAOG_03572, is the isoform above the exon skipped form? CAOG_07148, what is the isoform in the Filo stage just above the exon skipped isoform? CAOG_05784, What are the isoforms just below the inclusion isoform? Either the RT-PCR is not working as intended or there are potentially many other splice isoforms present*.

An extra band above the inclusion isoform is often observed when simultaneously amplifying two isoforms due to formation of a heteroduplex between one DNA strand from each isoform (these have long stretches of perfect sequence complementarity). Since the heteroduplex forms unpredictable secondary structures its formation varies between PCR reactions. In some cases it is also possible that unspecific bands are amplified in the PCR reaction. Nevertheless these effects do not impact the validation of the alternative splicing events in our study as we have confirmed the identity of the inclusion and exclusion isoforms by Sanger sequencing. We have added a note in the legend to acknowledge the presence of heteroduplexes in the RT-PCR reactions, and to confirm that the specific products have been confirmed by sequencing.

*4) There are several instances where the authors cite “significant enrichment of” an annotated function where they point to network diagrams presumably derived from cytoscape. As these are key conclusions in the paper, documentation of how these were derived as cytoscape analysis of pathway enrichment is not a standardized analysis. Data specifically noted are*
Figure 4
*and*
Figure 9*. My concern with these types of enrichment analyses are if the authors are specifically looking at genes they are interested in, or are annotated, versus actually testing against the null hypothesis. For example, are protein kinases actually more likely to have exon skipping than just any random set of genes in the genome (Blast2Go annotated or not)? What is the actual meaning of “genes with differential exon skipping are significantly enriched in protein kinase activity” and other related statements? These are key conclusions to this paper and as submitted it is difficult to evaluate if these statements are indeed true. Indeed, the authors claim, regarding protein kinases, that “[t]his represents the first example of a regulated exon network linked to a specific biological function in a unicellular organism”. This conclusion needs more documentation of how it was reached for me to agree*.

We apologize for the lack of clarity and we have made a significant effort to better explain the methods used. We tested GO enrichment by using Ontologizer, and not by “specifically looking at genes we were interested in”. What Ontologizer does is to test whether within a particular list of genes (in this case, genes that were found to be significantly overexpressed in one stage vs another) there is an overrepresentation of any GO (tested against the full list of genes and its associated GOs, the null hypothesis thus being that the proportion of genes for a specific GO category in the subset is a representation of the proportion of genes of these category in the complete list). We used the Topology-Weighted option to avoid having to apply a multiple hypothesis correction on the obtained p-values. Then, we used a p-value threshold of 0.01. The cytoscape networks, which seems to concern the reviewer as well, were built after the Ontologizer analysis had been done. That is, the network is only a visualization tool of the results obtained by Ontologizer (including number of genes in each GO, p-values, etc). Therefore, in response to answer the reviewer’s concern, protein kinases are significantly enriched for exon skipping when comparing any random set of genes in the genome.